# Comparison of EMT-Related and Multi-Drug Resistant Gene Expression, Extracellular Matrix Production, and Drug Sensitivity in NSCLC Spheroids Generated by Scaffold-Free and Scaffold-Based Methods

**DOI:** 10.3390/ijms232113306

**Published:** 2022-11-01

**Authors:** Xiaoli Qi, Alexandra V. Prokhorova, Alexander V. Mezentsev, Ningfei Shen, Alexander V. Trofimenko, Gleb I. Filkov, Rushan A. Sulimanov, Vladimir A. Makarov, Mikhail O. Durymanov

**Affiliations:** 1School of Biological and Medical Physics, Moscow Institute of Physics and Technology, National Research University, 141701 Dolgoprudny, Russia; 2Medical Informatics Laboratory, Yaroslav-the-Wise Novgorod State University, 173003 Veliky Novgorod, Russia

**Keywords:** tumor spheroids, extracellular matrix, cytotoxicity, anticancer drugs, 3D cell culture

## Abstract

Multicellular 3D tumor models are becoming a powerful tool for testing of novel drug products and personalized anticancer therapy. Tumor spheroids, a commonly used 3D multicellular tumor model, more closely reproduce the tumor microenvironment than conventional 2D cell cultures. It should be noted that spheroids can be produced using different techniques, which can be subdivided into scaffold-free (SF) and scaffold-based (SB) methods. However, it remains unclear, to what extent spheroid properties depend on the method of their generation. In this study, we aimed to carry out a head-to-head comparison of drug sensitivity and molecular expression profile in SF and SB spheroids along with a monolayer (2D) cell culture. Here, we produced non-small cell lung cancer (NSCLC) spheroids based on human lung adenocarcinoma cell line A549. Drug sensitivity analysis of the tested cell cultures to five different chemotherapeutics resulted in IC_50_ (A549-SB) > IC_50_ (A549-SF) > IC_50_ (A549-2D) trend. It was found that SF and SB A549 spheroids displayed elevated expression levels of epithelial-to-mesenchymal transition (EMT) markers and proteins associated with drug resistance compared with the monolayer A549 cell culture. Enhanced drug resistance of A549-SB spheroids can be a result of larger diameters and elevated deposition of extracellular matrix (ECM) that impairs drug penetration into spheroids. Thus, the choice of the spheroid production method can influence the properties of the generated 3D cell culture and their drug resistance. This fact should be considered for correct interpretation of drug testing results.

## 1. Introduction

Among different in vitro multicellular tumor models, spheroids are the simplest and one of the most widely used 3D cell cultures. Spheroids are tissue-like multicellular aggregates of spherical shape, consisting of dividing and non-dividing cells. Depending on the composition, spheroids can be homotypic, heterotypic or organotypic. Homotypic spheroids contain cells of the same type, while heterotypic spheroids contain different types of cells. Organotypic spheroids originate from different types of primary cells, for example, from isolated tumor cells [1].

Multicellular spheroids exhibit some hallmarks of tumor microenvironments, including the formation of nutrient gradients and a hypoxic core, a high number of cell-cell contacts, and deposition of extracellular matrix (ECM) by spheroid cells. Unlike regular monolayer cell cultures, 3D tumor spheroids more accurately reproduce the secretion of soluble bioactive molecules, gene expression patterns, and drug resistance [2,3]. For this reason, spheroids often serve as a testing tool for screening of novel drug candidates, studies of intercellular and intracellular signaling, evaluation of drug penetration, studies of cell-ECM interaction, and other applications [1,4,5].

There are multiple methods for spheroid production, which in general can be categorized into scaffold-free (SF) and scaffold-based (SB) techniques. Scaffold-based methods imply cell encapsulation into a polymeric scaffold of natural (collagen, Matrigel), semi-synthetic (PEGylated fibrinogen) or synthetic (poly-2-hydroxyethyl-methacrylate, polyvinyl alcohol) origin. Scaffold-free methods for generation of spheroids include cell centrifugation, placing the cells into a hanging drop, agitation of the cell suspension, and cultivation on non-adhesive surfaces [6]. It should be noted that multiple external parameters of tumor spheroids, including mean diameter, size uniformity, cell proliferation rate, and term of cultivation depend on the method of spheroid fabrication [7]. At the same time, it remains unclear whether the selection of spheroid generation technique affects internal spheroid properties such as viability, cell and ECM organization, protein expression pattern, and drug resistance. Therefore, the aim of this study was to compare these characteristics of non-small cell lung cancer (NSCLC) spheroids generated from one cell line (A549) using SF and SB techniques.

## 2. Results

### 2.1. Morphological Analysis of SF and SB A549 Spheroids

For generation of the SF A549 spheroids, we used a method of cell cultivation in non-adhesive agarose molds [8]. The use of this technique resulted in the formation of spheroids with spherical geometry in 2–3 days (Figure 1A). Despite an obvious increase in cell number and density, we did not find significant changes in spheroid size between day 3 and day 7 (Figure 1B). The spheroids reached the diameter of 115 ± 30 µm on day 7 (Table 1). In addition, the spheroids started to release single cells to the culture medium starting from day 4.

For generation of the SB A549 spheroids, the method of cell encapsulation into collagen I gel was used. Although the kit used is intended for production of tumor organoids according to manufacturer’s protocol (http://f-biomed.com/en/sanato-2/, accessed on 20 September 2022), we successfully used it for fabrication of SB A549 spheroids. This method resulted in large A549 spheroids, which were formed on days 2–3 after plating and acquired a non-spherical morphology (Figure 1C). Further incubation resulted in reductions in the spheroid size up to day 7 and the spheroids acquiring an ellipsoidal shape with an average aspect ratio of 1:2.5 (Figure 1C,D). The mean dimensions of the SB A549 spheroids on day 7 were 164 ± 62 μm (short axis) and 395 ± 84 μm (long axis) (Table 1).

Thus, both types of spheroids demonstrated similar formation kinetics. To evaluate the growth kinetics of SF and SB A549 spheroids, we measured the total ATP amount at different time points. The SF A549 spheroids reached the maximal ATP level on day 4 and it then decreased until day 7, whereas the ATP content in SB A549 spheroids reached the highest level on day 7 (Figure 2). These data indicate that growth of SF A549 spheroids stopped on day 4, while SB A549 spheroids exhibited continuous growth up to day 7.

### 2.2. Drug Resistance Analysis of A549 Cells in Monolayers, SF and SB A549 Spheroids

It was revealed in several studies that aggregation of cancer cells, including A549, into spheroids increases their resistance to anti-cancer drugs and nanotherapeutics [9,10,11]. Here, we observed the same phenomenon for A549 spheroids, which demonstrated enhanced drug resistance to five different chemotherapeutics in comparison with A549 cells grown in a monolayer (2D) (Figure 3A–E). In particular, the samples of SF A549 spheroids treated with pemetrexed (Figure 3F), gemcitabine (Figure 3G), and paclitaxel (Figure 3H) exhibited significantly higher IC_50_ values than the cells grown in 2D culture (*p* < 0.05). At the same time, the SB A549 spheroids turned out to be less sensitive to the drugs than their SF counterparts. The IC_50_ values were significantly higher for SB spheroids treated with cisplatin (Figure 3I), etoposide (Figure 3J), pemetrexed (Figure 3F), and gemcitabine (Figure 3G). Thus, in terms of resistance to chemotherapeutic drugs, we observed IC_50_ (SB) > IC_50_ (SF) > IC_50_ (2D) trend.

### 2.3. Expression of EMT Markers, Drug Resistance Enzymes, Cytokines, and Growth Factors in ML A549, SF and SB A549 Spheroids

Enhanced resistance of multicellular tumor spheroids to anticancer drugs in comparison with 2D cell cultures is usually attributed to impaired drug penetration and low oxygenation of the spheroid core. It is noteworthy that hypoxia along with cell-cell and cell-ECM interactions in 3D cultures influences the expression of genes involved in epithelial-to-mesenchymal transition (EMT), drug conversion and efflux, and pro-survival signaling. Therefore, the observed differences in drug resistance and viability between SF and SB A549 spheroids could be a result of altered expression of proteins involved in acquiring drug resistance.

As shown by RT-PCR analysis, the SB A549 spheroids exhibited a two-fold higher expression of two major pro-angiogenic molecules, vascular endothelial growth factor A (VEGF-A) and basic fibroblast growth factor (bFGF) (Figure 4A) than SF counterparts. Once the *VEGFA* gene is under the direct control of hypoxia-inducible transcriptional factor HIF-1α [12], an observed increase in VEGF-A expression indicates a higher level of hypoxia in SB A549 spheroids. It has been shown recently that cultivation in hypoxic conditions enhances interleukin-6 (IL-6) expression in A549 and H460 cell lines [13]. Here, we observed an almost two-fold increase in expression of this cytokine which contributes to the resistance of lung cancer cells to ionizing radiation and chemotherapeutics [14]. Interestingly, we also observed an increased expression of interferon γ (IFNγ) and interleukin-1β (IL-1β) as well as a two-fold upregulation of interleukin-23A (IL-23A) (Figure 4A). Although these cytokines do not seem to contribute to chemoresistance, they mediate immune suppression in the tumor microenvironment. For example, IL-1β, IL-6, IFN-γ and VEGF-A promote differentiation of myeloid progenitor cells to myeloid-derived suppressor cells (MDSCs), which inhibit killing activity of CD8+ T-cells [15]. As for IL-23A, this cytokine induces differentiation of pro-tumorigenic Treg cells [16].

Using western blotting, we analyzed the expression of the selected EMT markers, such as fibronectin, vimentin, α-smooth muscle actin (α-SMA), and drug resistance enzymes, including multidrug resistance protein 1 (MRP1) and glutathione S-transferase P1 (GSTP1) (Figure 4B–G). The obtained results demonstrated higher expression levels of the mentioned proteins, except for α-SMA, in spheroids compared with A549 cells in the 2D culture. Meanwhile, we did not find significant differences in expression of EMT markers and drug resistance enzymes between the SF and SB A549 spheroids.

### 2.4. Expression and Distribution of ECM Components in SF and SB Spheroids

It should be noted that penetration of chemotherapeutics is strongly limited due to overexpression of ECM components [17]. ECM as a part of tumor stroma significantly contribute to hindered transport of therapeutic agents through tumor interstitium [18,19]. According to previously published data [20], a significant part of NSCLC tumors is highly desmoplastic. In this regard, observed differences in drug resistance between SF and SB A549 spheroids could be explained by differences in ECM content and density. Using immunohistochemistry, we examined the ability of SF and SB A549 spheroids to produce different ECM components including collagen I, fibronectin, and laminin. We also studied the distribution of ECM proteins in the spheroid volume. It was found that all the proteins are produced in SF and SB A549 spheroids, although expression of ECM proteins was much higher in SB spheroids (Figure 5). Moreover, in SB spheroids all ECM proteins were distributed relatively uniformly throughout the spheroid volume and formed fibrillar networks with thick bundles (Figure 5A). In contrast, fibrous laminin and collagen I were deposited in the periphery of SF A549 spheroids, while fibronectin was organized into diffusive network.

Thus, enhanced chemoresistance of SB spheroids might be a result of higher volume and increased ECM deposition, which serves as a steric barrier to diffusion of therapeutic agents.

## 3. Discussion

The molecular expression profile and drug sensitivity of cancer cells strongly depend on the microenvironment. Assembly of the cells into 3D culture leads to an increase in intercellular contact number and cell-to-matrix interactions. Moreover, 3D multicellular models, including spheroids, reproduce the development of a hypoxic core and nutrient gradient that also affects gene expression, cell behavior, metabolism, and drug resistance [21]. The selection of a spheroid fabrication method should consider its advantages and limitations for a certain application. In particular, for drug screening analysis, the used method should provide spheroid uniformity in terms of mean diameter and shape. Otherwise, morphological heterogeneity of spheroids will be a source of data variability [22].

In this study we aimed to compare different characteristics of NSCLC spheroids generated from the A549 cell line using SF and SB techniques. In contrast to SF methods, SB techniques of spheroid fabrication are mainly based on the cell impregnation into a pre-assembled matrix that mechanically supports cancer cells and provides proliferative and pro-survival signaling via integrin receptors [21,23]. Moreover, this matrix might affect drug penetration and, therefore, spheroid cell viability under drug exposure. Thus, SF and SB spheroids could have different properties in terms of their molecular expression profile and chemoresistance.

Among multiple SF methods, we selected cell cultivation in non-adhesive agarose molds because this method results in a high number of uniform spheroids and has been previously used for generation of A549 spheroids [24,25,26]. We also obtained uniform spheroids of spherical shape and an average diameter of around 120 microns using this method (Figure 1A,B).

SB techniques are more difficult to perform, and they are rarely used for spheroid generation compared with SF methods. SB methods are mainly based on cell encapsulation into polymeric hydrogels [27,28], porous scaffolds [29], or ECM droplets surrounded a cell-impermeable shell [30,31]. We used here a commercially available kit for production of spheroids in collagen I gel droplets. This method requires a high number of cells per droplet and resulted in large ellipsoid spheroids with dimensions around 400 microns and 160 microns along the long and short axes, respectively (Figure 1C,D). Interestingly, direct measurement of spheroid size did not indicate the growth of tumor spheres (Figure 1). However, measurements of ATP levels demonstrated the growth of SF spheroids up to day 4 and further decreases in viability, whereas the growth rate of SB spheroids reached a plateau on day 7 (Figure 2). It should be noted that ATP level is a relevant measure of spheroid viability because the used reagent penetrates into spheroids of up to 650 μm in diameter [22].

The same method was applied for evaluation of A549 spheroid and monolayer cell culture sensitivity to five widely used for NSCLC treatment chemotherapeutics. It was found that SB spheroids were more resistant to therapeutic agents than their SF counterparts. In turn, all spheroid models exhibited enhanced chemoresistance in comparison with the 2D cell culture (Figure 3). The development of multidrug resistance in 3D multicellular spheroids can be a result of several processes. First, the cells in a monolayer are more available to the drugs than the cells in a spheroid. If drug accumulation in a spheroid is a diffusion-dependent process, the spheroids of larger size could be more resistant than their smaller counterparts, which has been shown, for example, in MCF-7 spheroids [32]. Second, the generation of 3D cell cultures leads to an increase in intercellular contacts, which contributes to upregulation of the EMT-related markers and chemoresistance proteins MDR1 and ATP binding cassette subfamily G member 2 (ABCG2), as was shown in the A549 spheroids [33]. Finally, increased drug resistance of spheroids could be a result of ECM expression, which contributes to pro-survival signaling and serves as a barrier to the diffusion of therapeutic molecules. Enhanced chemoresistance of ECM-rich MCF7 and OVCAR8 spheroids has been shown in comparison with their ECM-poor counterparts of the same size [34].

In part, enhanced drug resistance of SB A549 spheroids can be explained by their larger size. However, to better understand the observed differences, we evaluated the expression level of several EMT markers and proteins which mediate the development of chemoresistance. EMT is a biologic process that allows a polarized epithelial cell to assume a mesenchymal cell phenotype. EMT enhances cancer cell migratory capacity and increases their resistance to apoptosis. In addition, EMT upregulates the production of ECM proteins by cancer cells [35]. Elevated expression of EMT markers such as vimentin, fibronectin and α-SMA in spheroids (Figure 4B–E) indicates the onset of deep changes in the gene expression profile that can lead to increased drug resistance and other outcomes. Indeed, we observed upregulated GSTP1 and MRP1 in both SF and SB A549 spheroids compared with A549 cells cultured in a monolayer (Figure 4B,F,G). GSTP1 is crucial for the metabolism of cisplatin in vivo [36,37] and reduces sensitivity of cancer cells to etoposide [38]. Along with MRP1, GSTP1 is involved in neutralization of gemcitabine [39,40] and paclitaxel [41,42]. Therefore, our data suggest that higher expression of these proteins in spheroids could be an essential factor that improves their resistance to anti-cancer drugs. However, SB A549 spheroids are significantly more resistant to chemotherapeutics than their SF counterparts, whereas both types of spheroids express comparable amounts of MRP1 and GSTP1. To find an explanation for this phenomenon, we evaluated the cytokine and growth factor expression profiles as well as ECM production in SF and SB spheroids. It was found that SB A549 spheroids exhibit a two-fold higher expression of vascular endothelial growth factor A (*VEGFA*) that indicates a higher level of hypoxia in SB spheroids (Figure 4A). Besides *VEGFA* upregulation, hypoxic conditions lead to enhanced expression of the genes involved in chemoresistance, including anti-apoptotic *BCL2* gene, carbonic anhydrase 9 (*CA9*), multi-drug resistance 1 (MDR1) and others [43]. Moreover, hypoxia led to upregulation of IL-6 in SB spheroids (Figure 4A). This cytokine is involved in activation of pro-survival signaling pathways including Ras/MEK/ERK and PI3K/Akt [44] that could reduce drug sensitivity of SB spheroids.

Another important factor of the tumor microenvironment which contributes to chemoresistance is ECM. Despite the absence of pre-existing ECM, SF spheroids are able to produce ECM [45,46], which was also confirmed by our data (Figure 5B). At the same time, SF spheroids deposited collagen I and laminin on the periphery. In contrast, SB A549 spheroids produced much higher levels of fibronectin, laminin and collagen I, which were uniformly organized into a fibrous network (Figure 5A,C). It should be noted that the visualized ECM was produced by the cells in SB spheroids because pre-existing rat collagen I is not stained by the used antibodies.

Thus, enhanced chemoresistance of SB A549 spheroids can be a result of larger diameters, elevated levels of hypoxia and enhanced ECM production compared with their SF counterparts. It seems that the SB A549 spheroid better mimics the desmoplastic microenvironment of NSCLC tumors and helps to understand their enhanced drug resistance.

## 4. Materials and Methods

### 4.1. Cell Culture

Human adenocarcinoma alveolar epithelial cell line A549 was obtained from ATCC (CCL-185). The cells were cultured in DMEM/F12 medium supplemented with 10% FBS (*v*/*v*), 100 IU mL^−1^ penicillin, 100 µg mL^−1^ streptomycin and 2 mM Glutamax (Gibco) in a 95% humidified atmosphere containing 5% CO_2_ at 37 °C. For the experiments with monolayer (ML) cell cultures, the cells were seeded in 96-well flat-bottom plates (5000 cells per well) and incubated until 70% confluency, followed by drug treatment or harvesting to obtain the samples for Western blotting.

### 4.2. Generation of Scaffold-Free A549 Spheroids

SF A549 spheroids were generated using MicroTissues 3D Petri Dish^®^ micro-molds (Sigma-Aldrich, St. Louis, MO, USA) according to the manufacturer’s instructions. Briefly, the micro-molds were sterilized with anhydrous ethanol and allowed to dry under UV light for 30 min. Then, they were filled with sterile 2% (*w*/*v*) agarose solution, prepared in Milli-Q water. After solidification, the gelled agarose molds were released from the flexible 3D Petri Dish^®^ micro-molds and transferred to 6-well plates (3 agarose molds per well). To equilibrate the agarose gels, each well was filled with 1 mL of growth medium. After equilibration, the plates were placed in the incubator and incubated overnight.

Before seeding the cells, the medium was removed from both the culture plate and the molds. Then, 1.3 mL of fresh growth medium was added to each well and aliquots of 150 μL of growth medium with 40,000 cells were gently added into the molds. The medium was changed every other day. The size and morphology of the grown spheroids were examined by bright field microscopy using AxioVert.A1 microscope (Zeiss, Oberkochen, Germany) equipped with a Plan 10×/0.25 phase-contrast objective. On day 7, the spheroids were harvested for the following experiments.

### 4.3. Production of SB A549 Spheroids

For generation of the 3D spheroids, the wells of 96-well plates were coated with 50 μL of agarose solution (1% *w*/*v*) prepared in Milli-Q water and allowed to solidify in the incubator for 20–30 min. Next, 3.77 mg mL^−1^ collagen gel solution rat tail collagen type I (Corning, NY, USA) was diluted to 0.6 mg mL^−1^ using sterile Milli-Q water and 10X PBS. The pH was adjusted to 7.4 with 1N sodium hydroxide. Then, SANATO reagent (Phystech Biomed, Dolgoprudny, Russia) was added in a ratio 1:100 (*v*/*v*). After addition of A549 cells, 25 μL droplets containing 20,000 cells were plated. To solidify the gel, the plates were kept inside the incubator for 15–30 min. After solidification of the gel, the growth medium was added (100 μL per well), and the plates were returned to the incubator. The size and morphology of the spheroids were examined by bright field microscopy using an AxioVert.A1 microscope.

### 4.4. In Vitro Cell Viability Assay of Monolayered Cells and Spheroids

The viability of monolayered A549 cells and spheroids treated or non-treated with anti-cancer drugs was assessed using CellTiter-Glo^®^ 3D Cell Viability Assay (Promega, Madison, WI, USA) according to the manufacturers’ instructions. Briefly, the cells in monolayer or spheroids (10 SF spheroids per well or 1 SB spheroid per well) were exposed to different concentrations of cisplatin, etoposide, pemetrexed, paclitaxel or gemcitabine (all from Sigma-Aldrich, St. Louis, MO, USA) for 72 h. Incubation of the spheroids with drugs was carried out from day 4 to day 7. The final concentration of drugs in the wells was ranged from 1 to 200 μM. The untreated cells or spheroids served as a control. After incubation with drugs, equal volume of CellTiter-Glo^®^ 3D reagent was added to the wells. Next, the plate was vigorously shaken for 5 min and incubated for an additional 25 min at room temperature in the dark. The luminescence was measured using microplate reader CLARIOstar Plus (BMG LABTECH, Ortenberg, Germany).

### 4.5. Purification and Analysis of Total RNA and qPCR

RNA was purified with TRIZOL (Thermo Fisher Scientific, San Jose, CA, USA) as described earlier [47]. The prepared samples were subjected to spectral analysis. If the absorption ratio A260/A280 in at least one TRIZOL purified sample was lower than 2.0, the samples were repurified using CleanRNA Standard (Evrogen, Moscow, Russia). The integrity of the purified RNA was assessed electrophoretically in 1.5% agarose gel under non-denaturing conditions. The obtained RNA samples were converted to cDNA using the MMLV RT kit (Evrogen, Moscow, Russia). These samples were subjected to qPCR with the primers listed in Appendix A using CFX96 Real-Time PCR Detection System (Bio-Rad, Hercules, CA, USA). The 18S RNA gene was used as an endogenous control. The results were analyzed with the CFX Manager software supplied by the manufacturer. All samples were run in triplicate. Overall, three independent experiments were performed.

### 4.6. Western Blot

The cell or spheroid samples were lysed in 5% Triton X-100 buffer followed by centrifugation (13,000 rpm, 4 °C, 15 min) and collection of supernatants. After measurement the protein concentration using BCA Kit (Beyotime Institute of Biotechnology, Haimen, China), equal amounts of total protein (11 μg) were mixed with loading buffer containing β-mercaptoethanol (1:3) and boiled for 5 min. The proteins were separated by Laemmli electrophoresis and transferred to an Amersham^TM^ Hybond^TM^ 0.45 μm PVDF membrane (GE Healthcare, Chalfont St Giles, UK). The membrane was blocked with a blocking buffer (Bio-Rad, Hercules, CA, USA) for 1 h under gentle shaking and incubated for 2 h with primary antibodies. The following primary antibodies were used in the experiments: rabbit anti-alpha smooth muscle actin (ab12496, Abcam), rabbit anti-fibronectin (ab2413, Abcam), mouse anti-vimentin (ab8978, Abcam), rabbit anti-*MRP1* (ab260038, Abcam), rabbit anti-GSTP1 (AHP2471, Bio-Rad), rabbit anti-TGF beta 1 (ab92486, Abcam), and mouse anti-GAPDH (ab8245, Abcam). Then, the membrane was washed three times with TBS-T and incubated with secondary antibodies at room temperature for 1 h. Goat pAb to Ms IgG HRP-conjugated antibody (ab205719, Abcam) and Goat pAb to Rb IgG HRP-conjugated secondary antibody (ab205718, Abcam) were used with the primary antibodies produced in mice and rabbits, respectively. After incubation, the membrane was washed with TBS-T and deionized water. The protein bands were visualized using ChemiDoc XRS+ imaging system (Bio-Rad, Hercules, CA, USA) in chemiluminescence mode. The signal intensities were normalized to one of GAPDH. The quantification was performed using Quantity One software (Bio-Rad, Hercules, CA, USA).

### 4.7. Immunohistochemistry Analysis

After 6 days of culturing, the 3D spheroids were harvested, embedded into HistoPrep tissue embedding media (Thermo Fisher Scientific, San Jose, CA, USA), and frozen at −20 °C. Then, the frozen blocks were cut into 10 μm sections, fixed in a mixture of acetone and methanol (1:1) for 15 min and air-dried at room temperature. To identify the desired ECM proteins, the slides were washed with TBS-T (1x) and kept in blocking solution for 1 h at room temperature. Then, samples were incubated with primary antibodies for 3 h at room temperature. The following primary antibodies were used in the experiments: rabbit monoclonal anti-fibronectin primary antibodies (ab2413, Abcam), rabbit polyclonal anti-type I collagen antibodies (ab34710, Abcam), and rabbit polyclonal anti-laminin antibodies (ab11575, Abcam). After incubation with primary antibodies, the samples were washed with TBS-T and incubated with goat anti-rabbit IgG labeled with Alexa Fluor 488 (ab150077, Abcam) for 1 h at room temperature. Nuclear DNA was stained with DAPI for 10 min before the samples were covered with a cover glass. The images of spheroid sections were obtained using an inverted AxioVert.A1 microscope (Zeiss, Oberkochen, Germany) equipped with a ×20/0.6 objective lens.

### 4.8. Statistical Analysis

The experimental data analysis was carried out using Graphpad Prism 5 (GraphPad Software Inc., San Diego, CA, USA) software. The data are presented as mean ± standard deviation (SD). Each experiment was performed as minimum in triplicate. To determine statistical significance of differences between two groups, the nonparametric Mann–Whitney U-test was performed. The value *p* < 0.05 was considered to indicate a statistically significant difference.

## 5. Conclusions

In this study, we compared SF and SB A549 spheroids and analyzed their morphological characteristics, protein expression, ECM distribution, and drug resistance. We found that aggregation of A549 cells into SF or SB spheroids upregulates EMT markers and multidrug resistant genes. At the same time, SB A549 spheroids exhibited much higher resistance to five different chemotherapeutic drugs than their SF counterparts. This phenomenon can be attributed to their larger size, elevated hypoxia, and enhanced production of ECM proteins such as collagen I, fibronectin, and laminin. Thus, the SB A549 spheroid model can be used for mimicking desmoplastic NSCLC tumors, which display low sensitivity to chemotherapeutic agents.

It should be noted that we revealed here a IC_50_ (SB) > IC_50_ (SF) > IC_50_ (2D) trend in drug resistance behavior only in the A549 cell line, which is one of the most frequently used lung cancer models for in vitro studies. It cannot be excluded that drug resistance behavior could be different in spheroid models generated from other cell lines using SB and SF techniques. In any case, the method of spheroid production should be taken into account for drug resistance analysis of 3D multicellular models.

## Figures and Tables

**Figure 1 ijms-23-13306-f001:**
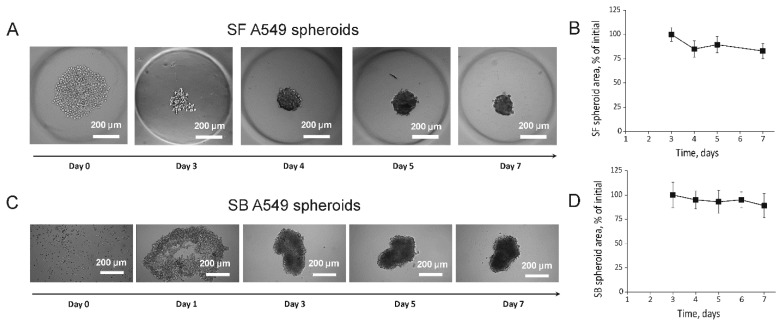
Morphological characteristics of SF and SB A549 spheroids. (**A**) Morphological changes in cultured SF spheroid over time. (**B**) Changes in average area occupied by SF spheroid during a one-week incubation. (**C**) Morphological changes in cultured SB A549 spheroids over time. (**D**) Changes in average area occupied by SF spheroid during a one-week incubation. (**B**,**D**) Data are shown as means ± SD (*n* = 8).

**Figure 2 ijms-23-13306-f002:**
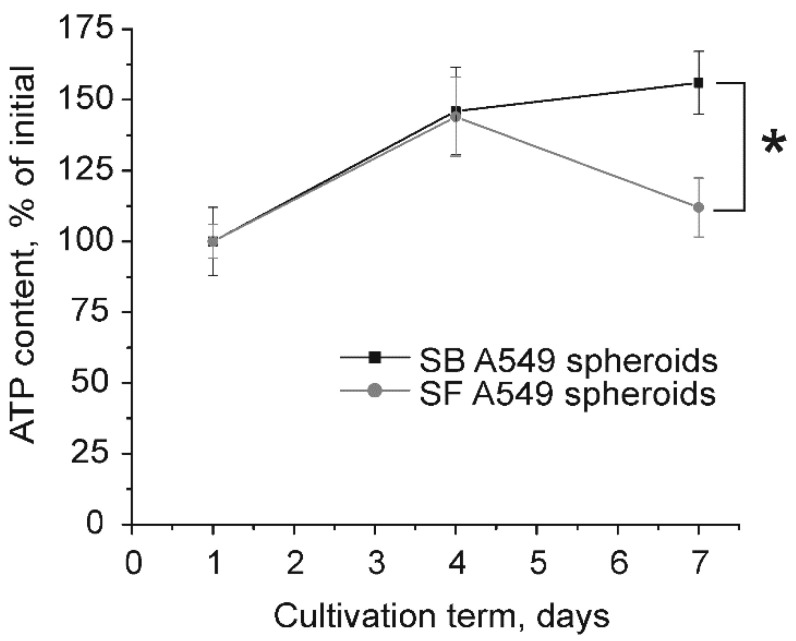
Measurement of total ATP amount in SF and SB A549 spheroids at different time points. ATP content correlates with viable cell number and reflects spheroid growth rate. Data are shown as means ± SD (*n* = 4–5). * *p* < 0.05 (Mann–Whitney U test).

**Figure 3 ijms-23-13306-f003:**
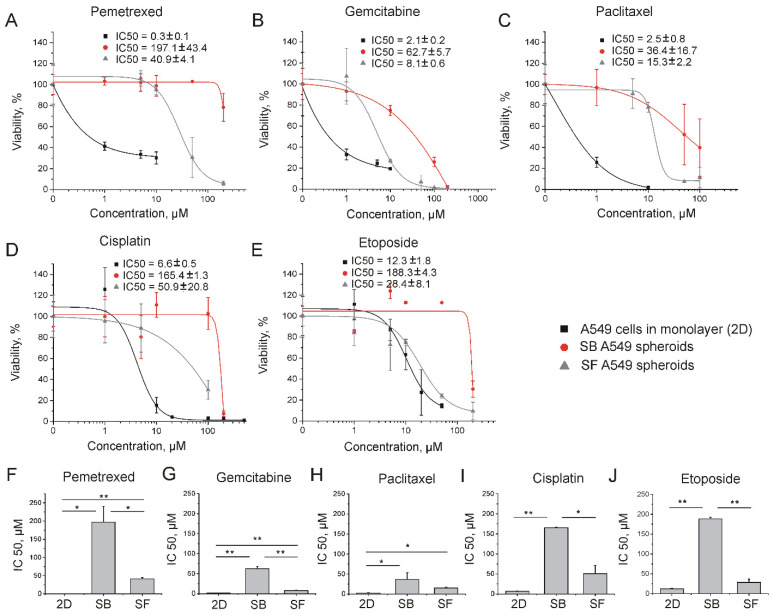
Analysis of drug resistance of different A549 cell cultures. Survival curves of A549 cells in monolayer culture (2D), SF and SB spheroids after 72 h of incubation with pemetrexed (**A**), gemcitabine (**B**), paclitaxel (**C**), cisplatin (**D**), and etoposide (**E**). Data are shown as means ± SD. Evaluation of IC_50_ values of ML A549 cells, SF and SB A549 spheroids after exposure to pemetrexed (**F**), gemcitabine (**G**), paclitaxel (**H**), cisplatin (**I**), and etoposide (**J**). Data are shown as means ± SD (*n* = 3). * *p* < 0.05, ** *p* < 0.01 (Mann–Whitney U test).

**Figure 4 ijms-23-13306-f004:**
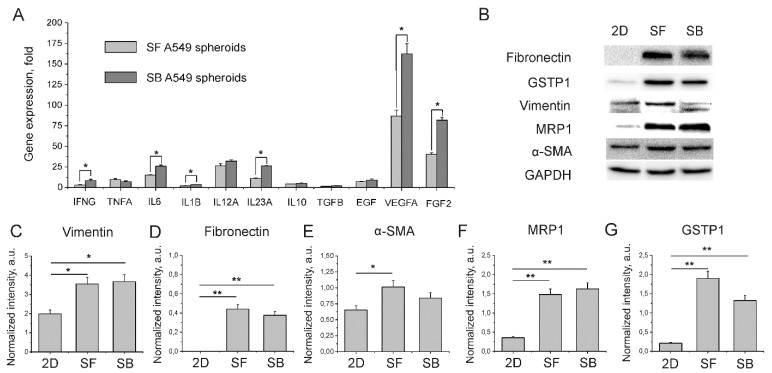
Expression of proteins contributing to chemoresistance and EMT of cancer cells in A549 monolayer culture (2D), SF and SB spheroids. (**A**) RT-PCR analysis of mRNA levels of growth factors and cytokines. (**B**) Analysis of protein expression by Western blot. Semiquantitative analysis of the expression levels of vimentin (**C**), fibronectin (**D**), α-SMA (**E**), MRP1 (**F**), and GSTP1 (**G**) in 2D A549 cell culture, SF and SB A549 spheroids. Data are shown as means ± SD (*n* = 3). * *p* < 0.05, ** *p* < 0.01 (Mann–Whitney U test).

**Figure 5 ijms-23-13306-f005:**
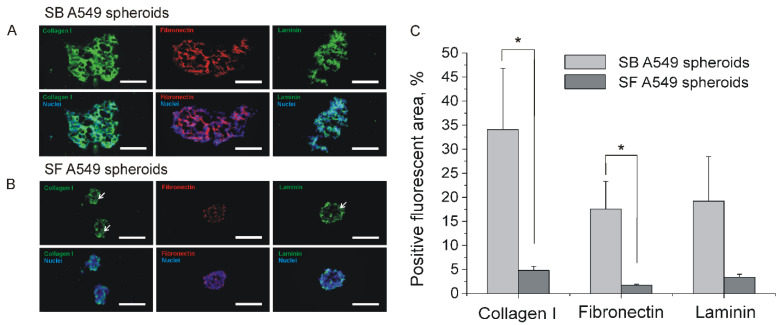
Distribution of ECM proteins in SF and SB A549 spheroids. (**A**) Images of SB A549 spheroid sections stained with antibodies against collagen I, fibronectin, and laminin. (**B**) Images of SF A549 spheroid sections stained with antibodies against ECM proteins. White arrows indicate peripheral deposition of collagen I and laminin. (**A**,**B**) Scale bar is 100 µm. (**C**) Quantification of ECM proteins deposited by SB and SF spheroids, determined as positive fluorescent area. Data are shown as means ± SD (*n* = 4–5). * *p* < 0.05 (Mann–Whitney U test).

**Table 1 ijms-23-13306-t001:** Characteristics of SB and SF A549 spheroids on day 7 *.

	Mean Diameter, µm	Mean Perimeter, µm	Cross-Aectional Area, mm^2^	Aspect Ratio
**SF spheroids**	115 ± 30	349 ± 45	(10.5 ± 0.4) × 10^−3^	1:1
**SB spheroids**	164 ± 62 (short axis) 395 ± 84 (long axis)	990 ± 115	(55.2 ± 3.5) × 10^−3^	1:2.5

* Data are shown as means ± SD (*n* = 8).

## Data Availability

Data will be made available on request.

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
