# Peer review of "Comparison of EMT-Related and Multi-Drug Resistant Gene Expression, Extracellular Matrix Production, and Drug Sensitivity in NSCLC Spheroids Generated by Scaffold-Free and Scaffold-Based Methods"

_ijms, 2022, doi:10.3390/ijms232113306_

Round 1

Reviewer 1 Report

In this manuscript by Qi Xiaoli (et al) the authors explore the differences in drug resistance to chemotherapy treatments in different cancer 3D spheroid models generated by scaffold (SB) vs scaffold free (SF) methods using a single lung cancer cell line A549. The authors use ATP consumption as a measure for cell viability in the presence and absence of panel of widely used chemotherapy drugs for NSCLC. They conclude that enhanced drug resistance seen in SB can be a result of higher diameter and elevated deposition of ECM, thus consideration has to be used when choosing the correct 3D model and results to drug testing. 

The paper on the whole is and referenced well and follows a logical progression, and would be of interest to the readership of this journal. 

I have a major concerns over the use of just single cancer cell line to base all the results from, and would advise them to at least repeat  a subset of experiments with other NSCLC see (https://www.atcc.org/products/tcp-2030) for panel of cell lines. I feel this would greatly improve the robustness of their results 

I have some other comments listed below 

major points 

1. figure 1 why not show other measurements of spheroids next to each in graph form as this is a bit lost in the text, such as mean dimensions, perimeter/area of the spheroid, some sort of spherical index( understand that the SB is not very spherical, but a measurement showing this would highlight this clearer.  

2.   

It is not clear in figure 3 at what time point the drugs were added it says incubated for 72h ,but  after how many days were spheroids grown for before addition of drugs? 

3.

figure 4 , I am not convinced by this blot for the P-glycoprotein that any meaningful quantitation could be drawn from this,  please could this be replaced with better example or data removed 

minor points 

1.

please state how many samples were mean calculated from in figure legends 

2.

 Line 73-74 “Further reinforcement of cell-cell interaction resulted in reduction of the spheroid size”

this is not clear what is mean by this and requires further explanation.

Author Response

Reviewer #1

In this manuscript by Qi Xiaoli (et al) the authors explore the differences in drug resistance to chemotherapy treatments in different cancer 3D spheroid models generated by scaffold (SB) vs scaffold free (SF) methods using a single lung cancer cell line A549. The authors use ATP consumption as a measure for cell viability in the presence and absence of panel of widely used chemotherapy drugs for NSCLC. They conclude that enhanced drug resistance seen in SB can be a result of higher diameter and elevated deposition of ECM, thus consideration has to be used when choosing the correct 3D model and results to drug testing.

The paper on the whole is and referenced well and follows a logical progression, and would be of interest to the readership of this journal.

I have a major concerns over the use of just single cancer cell line to base all the results from, and would advise them to at least repeat a subset of experiments with other NSCLC see (https://www.atcc.org/products/tcp-2030) for panel of cell lines. I feel this would greatly improve the robustness of their results.

Re: As mentioned in the Abstract and Introduction, the main goal of the study is to carry out head-to-head comparison of drug sensitivity and molecular expression profile in spheroids, generated by SF and SB methods. We revealed that these characteristics are different for SB and SF A549 spheroids and the choice of the spheroid generation technique can influence on the properties of the generated 3D cell culture and their drug resistance. Implementation of additional experiments with other NSCLC cell lines will neither doubt nor cancel the fact that was revealed based on SB and SF A549 spheroids as an example.

To improve the robustness of our data, we carried out an additional experiment with PCR analysis of growth factor and cytokine expression by SB and SF A549 spheroids. This experiment provided a valuable data for understanding of enhanced chemoresistance of SB spheroids (section 2.3, Figure 4A). In particular, SB A549 spheroids exhibited 2-fold higher expression of IL-6 and vascular endothelial growth factor A (VEGF-A) that indicates a higher level of hypoxia. Hypoxia is an important characteristic of NSCLC tumors, which leads to enhanced expression of the genes involved into chemoresistance, whereas IL-6 activates pro-survival signaling. These biomarkers could significantly contribute to reduced drug sensitivity of SB A549 spheroids as compared with SF counterparts. We added this information to Results (page 5, lines 15-21, page 6, lines 1-9) and Discussion (page 9, lines 9-14) sections.

I have some other comments listed below

major points

  1. figure 1 why not show other measurements of spheroids next to each in graph form as this is a bit lost in the text, such as mean dimensions, perimeter/area of the spheroid, some sort of spherical index (understand that the SB is not very spherical, but a measurement showing this would highlight this clearer).

Re: Parameters of the spheroids, which characterize spheroid size and geometry, were combined in Table 1 (page 3, lines 8-13). These parameters include 1) mean diameter (along 1 axis or two axes as in a case of SB), 2) spheroid perimeter, 3) spheroid area (in mm2), 4) mean aspect ratio (as an index of “sphericity”).

  1. It is not clear in figure 3 at what time point the drugs were added it says incubated for 72h, but after how many days were spheroids grown for before addition of drugs?

Re: We added this information to Materials and methods (page 10, line 27): “Incubation of the spheroids with drugs was carried out from day 4 to day 7.”

  1. figure 4, I am not convinced by this blot for the P-glycoprotein that any meaningful quantitation could be drawn from this, please could this be replaced with better example or data removed

Re: This blot is removed from the updated version of the manuscript.

minor points

  1. please state how many samples were mean calculated from in figure legends

Re: In the section “4.8. Statistical analysis” we stated that “Each experiment was performed as minimum in triplicate.” We added the exact number of repetitions to all figure captions, where necessary.

  1. Line 73-74 “Further reinforcement of cell-cell interaction resulted in reduction of the spheroid size”.

this is not clear what is mean by this and requires further explanation.

Re: This statement is reworded un the updated manuscript (page 2, lines 41-42): “Further incubation resulted in reduction of the spheroid size up to day 7 and acquirement of ellipsoidal shape with average aspect ratio of 1:2.5 (Figure 1C,D).”

Reviewer 2 Report

The manuscript by Qi et al compares EMT markers, genes, ECM and drug sensitivity in spheroids generated by scaffold free and scaffold based methods. 

Its an interesting study and I have minor concerns.

Fig 1: Scale bar is missing, why the growth was monitored for 7 days only? How about the viability of these models?

Fig 2: Why ATP in SF was lower than SB at dat 7? Not clear.

Fig 3: Resolution of figures could be improved. Please indicate IC50 in graph as well. WHy different drugs reacted differently to SF and SB?

Fig 4: ECM proteins remained the same essentially between SF and SB. So what is conclusion of this result? 

Line 216: Hoe does elevation of ECM markers lead to enhanced drug resistance? Is it a direct effect?

Line 235: diameters of SF and SB are almost similar. So, how can diameter contribute to enhanced chemoresistance?

SF is more resistant than SB. What is mechanism?

Author Response

Reviewer #2.

The manuscript by Qi et al compares EMT markers, genes, ECM and drug sensitivity in spheroids generated by scaffold free and scaffold based methods.

Its an interesting study and I have minor concerns.

(1) Fig 1: Scale bar is missing, why the growth was monitored for 7 days only? How about the viability of these models?

Re: The scale bar has been added to the image. During the measurement of ATP level (Figure 2), we observed that the growth of SF spheroids peaks at day 4 and falls further as viability decreases, while growth rate of SB spheroids reaches a plateau at day 7. While further incubation leads to decrease in viability, we have worked within 7-day timeframe. The viability of these models was described in the text (page 3, lines 16-19): “SF A549 spheroids demonstrated reaching the maximal ATP level on day 4 and further decrease on day 7, whereas ATP content in scaffold-based (SB) A549 spheroids achieved the highest level on day 7 (Figure 2). These data indicate that growth of SF A549 spheroids stopped on day 4, while SB A549 spheroids exhibited continuous growth up to day 7.”

(2) Fig 2: Why ATP in SF was lower than SB at dat 7? Not clear.

Re: Fig. 2 shows the relative ATP content as a percent of the initial amount. Hence, the starting amount was set as 100 %. We did not compare the absolute ATP amount because it was a different number of cells per spheroid. However, comparison of the relative ATP content over time indicated higher viability of SB spheroids on day 7, as compared with SF counterparts. Most probably, it can be explained by activation of proliferative and pro-survival signaling via integrin receptors from pre-assembled matrix in SB spheroids.

(3) Fig 3: Resolution of figures could be improved. Please indicate IC50 in graph as well. WHy different drugs reacted differently to SF and SB?

Re: We increased the resolution of all figures. IC50 values are added. As for differences in drug sensitivity of SB and SF, it can be a result of several factors. First, SB spheroids are larger than SF spheroids. Increase in diameter could contribute to chemoresistance because small-molecule drugs penetrate tumor tissue mainly due to passive diffusion. Therefore, drug concentration in a central core will be a function of time and distance. In one’s turn, distance depends on a spheroid size. As a result, less amount of medicines reaches a central part of the spheroid. Second, increase in spheroid diameter leads to decreased concentration of oxygen in a central core caused by limited diffusion. Development of hypoxia significantly contributes to chemoresistance via different mechanisms including IL-6-mediated activation of pro-survival signaling. We carried out RT-PCR analysis, as an additional experiment, and detected 2-fold increase in IL-6 and VEGF-A expression in SB spheroids that indicates enhanced level of hypoxia in them as compared with SF counterparts. Third, SB spheroids produce higher amount of ECM. Elevated ECM content impedes penetration of small-molecule medicines. Moreover, integrin receptors interact with ECM components and transmit pro-survival and proliferative signals to the cells, making them more resistant to the drugs.

(4) Fig 4: ECM proteins remained the same essentially between SF and SB. So what is conclusion of this result?

Re: Yes, we agree that all types of ECM proteins are present in both types of spheroids. However, quantitative analysis showed a marked increase in positive fluorescence area of ECM proteins in SB spheroids as compared with SF counterparts. In addition, in SF spheroids ECM was deposited mainly on a periphery, whereas in SB spheroids we observed a uniform ECM distribution. Elevated and uniform production of ECM can contribute to enhanced chemoresistance of SB spheroids as mentioned in the Discussion part (page 7, lines 27-30, page 8, lines 1-3).

(5) Line 216: Hoe does elevation of ECM markers lead to enhanced drug resistance? Is it a direct effect?

Re: First, the extracellular matrix (ECM) acts as a steric barrier, which impedes penetration of small-molecule medicines. Second, integrin receptors on the cell surface interact with ECM components and transmit the pro-survival and proliferative signals to the cells, making them more resistant to the drugs. We mentioned both points with references in the Discussion part (page 7, lines 27-30, page 8, lines 1-3).

(6) Line 235: diameters of SF and SB are almost similar. So, how can diameter contribute to enhanced chemoresistance?

Re: We listed geometrical parameters of the spheroids in Table 1. SB spheroids exhibited higher size than SF spheroids. Increased diameter could contribute to chemoresistance because of 2 possible reasons. First, small-molecule drugs penetrate tumor tissue mainly due to passive diffusion. Therefore, drug concentration in a central core will be a function of time and distance. In one’s turn, distance depends on a spheroid size. As a result, less amount of medicines reaches a central part of the spheroid. Second, increase in spheroid diameter leads to decreased concentration of oxygen in a central core caused by limited diffusion. Development of hypoxia significantly contributes to chemoresistance via different mechanisms including IL-6-mediated activation of pro-survival signaling. We carried out RT-PCR analysis, as an additional experiment, and detected 2-fold increase in IL-6 and VEGF-A expression in SB spheroids that indicates enhanced level of hypoxia in them as compared with SF counterparts. We added this information to Results (page 5, lines 15-21, page 6, lines 1-9) and Discussion (page 9, lines 9-14) sections.

(7) SF is more resistant than SB. What is mechanism?

Re: Please, see our response on the comment (3).

Round 2

Reviewer 1 Report

The authors have done well in addressing  most of my points raised and thank them for their efforts. 

“Implementation of additional experiments with other NSCLC cell lines will neither doubt nor cancel the fact that was revealed based on SB and SF A549 spheroids as an example.”

Re: 

I still have concerns over just the use one cancer cell line, to understand general mechanisms to drug resistance in SB and SF NSCLC spheroids, I understand that doing these experiments in other cell lines will not cancel the results presented here for the A549 spheroids. 

If this work was repeated in different cell lines and found to be the same then this would greatly support this data. 

If however a contradictory result was found from another cell line this would put the explanations to the mechanism into some doubt therefore I do not agree with the authors comments on this part. 

I would find an acceptable solution to this disagreement  if the authors were to highlighted in the text this single cell line caveat clearly and hope the authors design of future experiments going forward, will consider my suggestions.  

Author Response

Reviewer #1

The authors have done well in addressing  most of my points raised and thank them for their efforts. 

“Implementation of additional experiments with other NSCLC cell lines will neither doubt nor cancel the fact that was revealed based on SB and SF A549 spheroids as an example.”

Re: 

I still have concerns over just the use one cancer cell line, to understand general mechanisms to drug resistance in SB and SF NSCLC spheroids, I understand that doing these experiments in other cell lines will not cancel the results presented here for the A549 spheroids. 

If this work was repeated in different cell lines and found to be the same then this would greatly support this data. 

If however a contradictory result was found from another cell line this would put the explanations to the mechanism into some doubt therefore I do not agree with the authors comments on this part. 

I would find an acceptable solution to this disagreement  if the authors were to highlighted in the text this single cell line caveat clearly and hope the authors design of future experiments going forward, will consider my suggestions.  

Re: We have addressed this point in an updated version of the manuscript in the Conclusions section (page 12, lines 15-20): “It should be noted that we revealed here a IC50 (SB) > IC50 (SF) > IC50 (2D) trend of drug resistance behavior only in A549 cell line, which is one of the most frequently used lung cancer models for in vitro studies. It cannot be excluded that drug resistance behavior could be different in spheroid models generated from other cell lines using SB and SF techniques. Anyway, the method of spheroid production should be taken into account for drug resistance analysis of 3D multicellular models.”

Research in a field of multicellular 3D cancer models and biomaterials is the main focus of our group. Of course, we plan further studies in this field aiming at analysis of gene expression alterations in 3D versus 2D cell cultures for different cancer models.